# Effect of EC Levels of Nutrient Solution on Glasswort (*Salicornia perennis* Mill.) Production in Floating System

Esra Okudur [1,*] and Yuksel Tuzel [2]

1   Graduate School of Natural and Applied Sciences, Ege University, Bornova, Izmir 35100, Türkiye
2   Department of Horticulture, Faculty of Agriculture, Ege University, Bornova, Izmir 35100, Türkiye
*   Correspondence: esra.okudur@batman.edu.tr

**Abstract:** Halophytes have evolved to tolerate high salinity environments. The halophyte glasswort (Salicornia and Sarcocornia species) grows by the sea or in salty soils and can be consumed with pleasure. In this study, the cultivation of glasswort was studied by testing the effects of different electrical conductivity (EC) levels (10, 15, 20, 25, 30 and 35 mS cm$^{-1}$) of a nutrient solution. *Salicornia perennis* Mill. was grown on floating systems in unheated greenhouse conditions. To adjust the different EC levels, sodium chloride was added to the Hoagland nutrient solution (EC: 2 mS cm$^{-1}$). Plant growth and yield parameters, shoot color, evapotranspiration, and shoot nutrient content were determined. Among the tested EC levels, the highest plant height (33.56 cm), shoot (172.75 g) and root fresh weights (41.74 g), stem diameter (7.85 mm), and fresh biomass (2864.06 g m$^{-2}$) were obtained from an EC level of 25 mS cm$^{-1}$. There were no significant differences in shoot color excluding b* and chroma values. It was concluded that glasswort could be grown in hydroponic systems as a new crop and that an EC value of 25 mS cm$^{-1}$ is the most appropriate for the cultivation of *Salicornia perennis* Mill. on floating systems.

**Keywords:** soilless culture; salinity; halophyte; hydroponic





## 1. Introduction

The world population is projected to reach 9.7 billion in 2050 [1] and this consequently will result in a food increase need of at least 70% [2]. Increasing population and food demands have exerted great pressure on agricultural production and the availability of resources. In addition, the impact of climate change on agriculture is evident and well known [3].

In recent years, salinity is one of the most common and critical problems threatening global food security and environmental sustainability [4]. Saline areas have been increasing by 10% annually due to various reasons [5,6]. It is predicted that 50% of all arable land will be affected by salinity by 2050 [7].

Salinization causes the loss of soil fertility and desertification and negatively affects vegetation and beneficial microorganisms [8]. Salt stress impairs plant growth due to water stress, excessive intake of elements such as sodium (Na$^+$) and chlorine (Cl$^-$) and imbalance in nutrient intake. Likewise, typically with salinity, oxidative stress occurs due to the generation of reactive oxygen species [9,10]. Different methods, such as the use of amendments, appropriate irrigation, drainage and land-use strategies, cultivation of tolerant genotypes, conservation agriculture, phytoremediation, and bioremediation, are techniques that have been used to reclaim the soils and mitigate the effects of salinity [4].

Studies on the use of salt-resistant plants against salinity have gained more prominence in recent years. Halophytes are considered to be one of the best plants since they have the potential of salt-responsive genes and promoters [11]. Research on halophytes can contribute to (i) garnering more information concerning the mechanisms of survival and maintaining productivity in salt water, which subsequently could be practical in developing

tolerant varieties of traditional crops, (ii) evaluating the long-term feasibility of agriculture in high salinity, and (iii) becoming a direct source of potential new crops [12]. Halophytes are commercial alternatives to traditional agricultural products and can be used as fodder, energy, oilseed, and protein crops. It is predicted that halophytes may be an important component of the agricultural system in the 21st century [13].

Halophytes have evolved to tolerate high salinity environments; they are plants that survive by growing within a salt concentration of 200 mM NaCl or higher and constitute approximately 1% of the world's flora. The salinity tolerance of halophytes is based on the uptake and distribution of $Na^+$, $K^+$ and $Cl^-$ and the synthesis of osmoprotectants or organic compatible solutes [14]. Glasswort (Salicornia and Sarcocornia species), which is one of the most important halophyte genera in the world today, is considered to be one of the most salt-resistant plants growing in salty regions [15]. It can tolerate 1000 mM or more of NaCl [16].

Salicornia's name is derived from the Latin word for "salt" and the genus Salicornia is represented by approximately 25 to 30 species worldwide [17]. Sarcocornia is a halophytic, edible, succulent plant belonging to the Amaranthaceae family. To date, it is comprised of 28 species in saline environments worldwide; its morphological and taxonomic properties are similar to Salicornia, which is well known and has a commercial value [18]. The succulent shoots of species belonging to both genera have gained commercial importance under various names. Sarcocornia species, unlike Salicornia, are perennials, meaning they can be harvested throughout the year. Therefore, Sarcocornia species are considered promising gourmet vegetables to explore in the context of climate change, soil and water salinization, and eco-sustainability [19]. The origin of *Sarcocornia perennis* Mill. is Atlantic and Mediterranean coasts in West and South Europe and North Africa [20].

The species Salicornia and Sarcocornia have the potential of usage for multiple purposes (such as food, pharmaceuticals, bioethanol production, biofilter, and phytoremediation). The oriental pharmacopoeia has reported some medicinal uses for the treatment of oxidative stress (such as asthma and diabetes); as a source of vitamin A, minerals, and fatty acids; and as a vegetable. Some studies have also demonstrated its benefits as an edible species in terms of nutritional properties [21]. It is reported that the Salicornia plant contains high levels of ascorbic and dehydroascorbic acids (<100 mg 100 $g^{-1}$) and carotenoids (5 mg 100 $g^{-1}$) [15,22]. Salicornia species are promising as functional foods considering their high nutritional value in terms of mineral compounds, including Mg, Na, Ca, Fe, and K, and many bioactive compounds such as phytosterols, polysaccharides, and phenolic compounds, especially flavonoids and phenols [23]. A study conducted with *Sarcocornia perennis* Mill. showed that total antioxidant capacities were found to be significant with high antimicrobial activity potential [24].

Although the various benefits of Salicornia and Sarcocornia species are known, scientific research on these plants is still very limited. Studies were conducted focusing on multiple factors, such as those affecting growing conditions, yield, nutritional values, day length [25], varying salt levels [26–29] and irrigation with sea water [30–34] or salty water [35–37] as well as the effects of the harvest period and storage conditions [38,39] on the nutritional profile [40–42] and plant anatomical characteristics [43]. The mapping of suitable areas for cultivation [44], methods of monitoring the salinity adaptation feature [45], and the soil improvement effect were also studied and comprehensive reviews were prepared [15,20,21,32,46–48].

These species can be grown in insulated wetlands and irrigated with sewage water with high salinity or effluents from aquaculture to increase sustainability [49]. Additionally, the cultivation period can be controlled. It is reported that the yield (harvested biomass) is higher in hydroponic systems, but there is a need for optimization in the nutrient solution [50]. In this study, we studied *Salicornia perennis* Mill., which is one of the halophytes living in salty areas [51] and aimed to grow this species on floating systems and to test different salinity levels of nutrient solution on plant growth and yield parameters, shoot color, evapotranspiration, and shoot nutrient content.

## 2. Materials and Methods

### 2.1. Test Site, Plant Material and Growing Conditions

This research was carried out in an unheated polycarbonate greenhouse (528 m$^2$) located at the Batı Raman Campus of Batman University (37°47′22″ N, 41°03′55″ E, altitude 550 m) during the spring season of 2022.

The plant material was collected from the Antalya province (Türkiye) (36°50′32″ N, 31°08′44″ E, altitude 0 m). The EC and pH values of the site were 18.4 mS cm$^{-1}$ and 7.88, respectively. The species was identified by Prof. Dr. Ahmet Emre Yaprak (Ankara University) as *Salicornia perennis* Mill. (formerly *Sarcocornia perennis* Mill.). The average height of cuttings used in this study was 7 cm with 0.9 g average weight.

A total of 6 floating systems were used in the experiment. Each system was composed of ponds (113 cm × 300 cm × 25 cm) covered with polyethylene foil. Styrofoam with a thickness of 20 mm was placed on the solution to float the plants. Based on the daily measurements via an oxygen meter, oxygen was provided to the plant roots with an air motor. In the study, the average amount of dissolved oxygen applied to the solution was 6.4 mg L$^{-1}$.

The plants were placed at a distance of 15 cm between the rows and 20 cm on the row with a plant density of 17.7 plants per m$^2$. The experiment was conducted once and each replication had 20 plants with 60 plants having 3.39 m$^2$ in each treatment. On 21 February 2022, the cuttings were placed in plastic pots with a diameter of 5 cm and with holes at the bottom and side surfaces. Plants were harvested once 15 weeks later on 6 June 2022.

Tunnels were constructed with the same width of ponds and 110 cm height since the wind had thrown the polycarbonates off the roof before the planting. (Figure 1). Temperatures within the tunnels varied between 7.6 and 21.8 °C with an average of 15.2 °C. The temperature of the nutrient solution was an average of 16.6 °C.

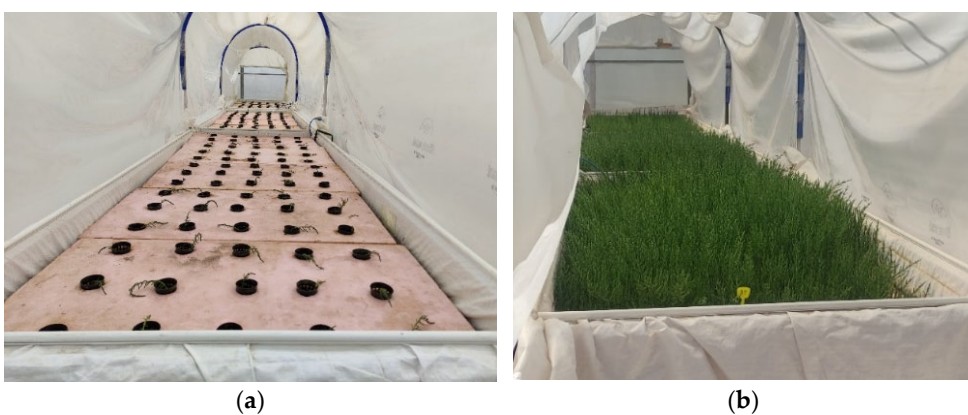

|                |                |
| :------------: | :------------: |
| (**a**)        | (**b**)        |

**Figure 1.** Plants on the first day (**a**) and harvesting date (**b**).

### 2.2. Nutrient Solution and Treatments

The water and nutrient requirements of the plants were covered with a complete nutrient solution prepared according to the results of the water analysis (Table 1). A nutrient solution of Hoagland and Arnon [52] was used: (mg L$^{-1}$) N 210, P 31, K 234, Ca 160, Mg 34, S 64, Fe 2.5, Mn 0.5, Zn 0.05, Cu 0.02, B 0.5 and Mo 0.01 with an electrical conductivity of (EC) 2 mS cm$^{-1}$ and a pH of 8.34.

Six EC levels of nutrient solution were tested. Salinity levels were adjusted by adding sodium chloride into the Hoagland nutrient solution as 10, 15, 20, 25, 30, and 35 mS cm$^{-1}$ (Figure 2). The nutrient solution was not replaced and/or not completed during the growing period. The differences in EC levels during the study were EC: 10 + 0.6 mS cm$^{-1}$, EC: 15 + 0.63 mS cm$^{-1}$, EC: 20 + 0.67 mS cm$^{-1}$, EC: 25 + 1.01 mS cm$^{-1}$, EC: 30 + 0.89 mS cm$^{-1}$, and EC: 35 + 0.97 mS cm$^{-1}$.

**Table 1.** Water analysis.

| Ions | Value | Ions | Value |
|---|---|---|---|
| Nitrate ($NO_3^-$) (mg $L^{-1}$) | 14.69 | Iron (Fe) (µg $L^{-1}$) | - |
| Potassium (K) (mg $L^{-1}$) | 0.75 | Mangane (Mn) (µg $L^{-1}$) | - |
| Phosphorus (P) (mg $L^{-1}$) | 0.045 | Zinc (Zn) (µg $L^{-1}$) | 13.51 |
| Calcium (Ca) (mg $L^{-1}$) | 44.89 | Copper (Cu) (µg $L^{-1}$) | 3.098 |
| Magnesium (Mg) (mg $L^{-1}$) | 29.17 | Borium (B) (µg $L^{-1}$) | 12.15 |
| Sodium (Na) (mg $L^{-1}$) | 0.75 | Fluoride ($F^-$) (mg $L^{-1}$) | 0.30 |
| Chlor (Cl) (mg $L^{-1}$) | 7.93 | Sulphate ($SO_4^{-2}$) (mg $L^{-1}$) | 37.05 |

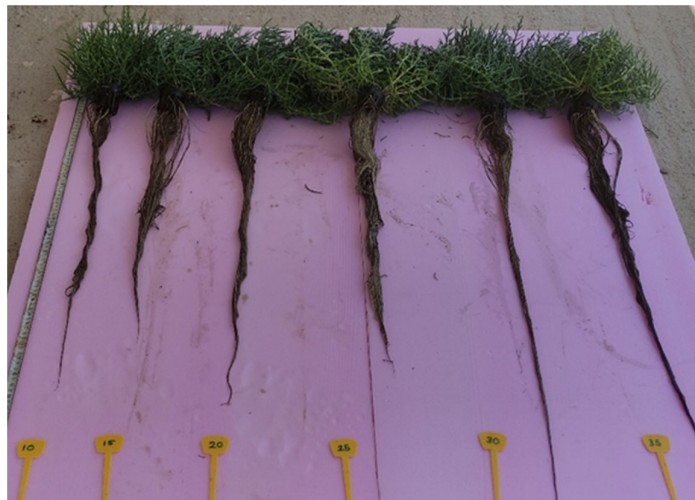

**Figure 2.** Glasswort plants at different EC levels (10, 15, 20, 25, 30, and 35 mS $cm^{-1}$).

*2.3. Measurements and Calculations*

At harvest, ten plants from each replicate of treatments were randomly selected for plant growth measurements. Plant height from the Styrofoam surface to the top of the plants was measured with a ruler and the results were expressed in cm. For the root length, plants were cut 1 cm above the root collar and measured with a ruler until the root tip, and the results were recorded in cm. Shoots and roots were weighed for fresh weight (g) and then dried in a thermos-ventilated oven at 65 °C and weighed for the dry weight (g). Stem diameter was indicated in millimeters (mm) by measuring with a digital caliper. The shoot-to-root ratio was calculated by dividing the fresh shoot weight by the fresh root weight. Fresh biomass was determined as g $m^{-2}$ by weighing the fresh weights of all plants in the experiment, and the results were calculated over $m^2$. Shoot color values were measured with a colorimeter (NR10QC Precision Colorimeter, China) and a mean L*, a*, b*, chroma and hue angle values were determined. The lightness (L*) value determines the brightness, the a* value determines the green color, the b* value determines the yellow color, the chroma (C*) value determines the saturation of the color and the hue (h°) value determines the angle of the color. The chroma [$C* = (a*^2 + b*^2)^{\frac{1}{2}}$] and hue angle [$h° = arctan(b*/a*)$] were also calculated.

In the study, evapotranspiration (ET) was calculated by subtracting the amount of solution on the last day from the amount on the first day (847.5 L) and the results were expressed as L $plant^{-1}$. Water use efficiency (WUE) was calculated as the ratio of fresh biomass to water consumption [53].

For mineral analysis, shoots of each treatment were dried in a Memmert UN160 oven for 24 h at 70 °C. The dried samples were ground with a Spice & Herb Grinder/IC-10B brand/model herb grinder and passed to sieves. The concentrations of Mg, Zn, Mn, Fe, and P were determined from nitric acid digestions (Berghof Speedwave MWS-2 Microwave Pressure Digestion) and ICP-OES (Plasma optical emission spectrometry) analyses. K, Ca,

and Na analyses were performed with a Flame photometer (Perkin Elmer ICP-OES Optima 2100 DV. and Jenway PFP7). Chloride analysis was determined according to the Mohr salt determination method [54].

### 2.4. Experimental Design and Statistical Analysis

The experimental design was randomized parcels with three replicates. To determine any statistically significant differences, data were subjected to analysis of variance by using an SPSS 17.0 package program. Duncan's Multiple Range Test at a 5% significance level ($p < 0.05$) was used to determine the differences between the means.

## 3. Results and Discussion

### 3.1. Plant Growth

It was demonstrated that *Salicornia perennis* Mill., a common species along the western and southern coastline of Türkiye [55], could be grown successfully in hydroponics. However, the effects of different EC levels of nutrient solution on measured plant growth parameters were observed to be significant (Figure 3). Plant height varied between 26.93 and 33.56 cm. It was the highest at EC levels of 20 and 25 mS cm$^{-1}$ with an average of 33.3 cm followed by 35 and 30 mS cm$^{-1}$. Plant height was lower at 19.5, 19.8, 1.6, 10 and 5.1% at 10, 15, 20, 30 and 35 mS cm$^{-1}$, respectively compared with 25 mS cm$^{-1}$ having the highest value (Figure 3A).

Shoot fresh weights were the lowest at EC levels of 10 and 15 mS cm$^{-1}$. Shoot fresh weight at 25 mS cm$^{-1}$ was higher at 145.4, 90.1, 9.9, 22.3 and 8.5% than at 10, 15, 20, 30 and 35 mS cm$^{-1}$, respectively (Figure 3B). Shoot dry weight was higher at 25, 30 and 35 mS cm$^{-1}$ compared to the lower EC levels (Figure 3C). Stem diameter changed between 6.1 mm (the lowest in 10 mS cm$^{-1}$) and 7.85 mm (the highest in 25 mS cm$^{-1}$) (Figure 3D).

Root length was the highest (94.9 cm) at the EC level of 30 mS cm$^{-1}$ and it was 1.32, 1.60, 1.27, 1.31 and 1.14 times higher than at 10, 15, 20, 25 and 35 mS cm$^{-1}$ (Figure 3E). Root fresh weight was the highest at 25 mS cm$^{-1}$ while the highest root dry weight was at 35 mS cm$^{-1}$ (Figure 3F,G). Root morphological features of halophytes are affected by salinity [56,57] and increasing salinity may increase the root dry weight and length and modify root anatomy due to the adaption mechanism of species; however, the response may change with the genotypes and nutrition application [58].

Shoot: root ratio gradually increased with increasing salinity [31] and the highest ratio (5.41) was at 35 mS cm$^{-1}$. Shoot: root ratio at 10, 15, 20, 25 and 30 mS cm$^{-1}$ was lower at 46.95, 37.34, 17.93, 15.53, and 5.73%, respectively (Figure 3H).

In the natural habitats of Salicornia species, soil salinity of the samples obtained from the sites was predominated by *Salicornia europaea* ranging from 18.9 to 142.8 mS cm$^{-1}$ at Rittman, Ohio [59], which indicates the presence of Na and Cl in the Salicornia zone [60], 63.1 and 38.5 mS cm$^{-1}$ in the industry area and natural saline site in Poland, respectively [46], and 70 ± 6 mS cm$^{-1}$ in unprotected and open grazing area in Adana, Türkiye [61]. In the pot experiments, testing different seawater and/or salinity treatments (25, 50, 75, and 100% seawater), it was determined that the response of plants showed differences in growing stages, in particular, at germination and with ecotypes [29,31,46,50].

In this experiment, plant height, shoot fresh and dry weights, root fresh weight, and stem diameter were the highest at 25 mS cm$^{-1}$. In all measured parameters in terms of plant growth, there were lower values at the lower EC levels (10 and 15 mS cm$^{-1}$) (Figure 3A–H). Araus et al. [29] compared three salinity levels, namely fresh water (0.3 mS cm$^{-1}$), brackish water (25 mS cm$^{-1}$), and seawater (40 mS cm$^{-1}$) and the highest seed weight, shoot biomass, plant height, and branch number of two *S. europaea* ecotypes were obtained from the brackish water having an EC level of 25 mS cm$^{-1}$. These results reveal that plant growth performance is higher at moderate salinity levels around 25 mS cm$^{-1}$; however, the plant response yields variations in different species, ecotypes [62,63], growing conditions [25,64], plant density [65], harvest regime [25], and nutrition [58].

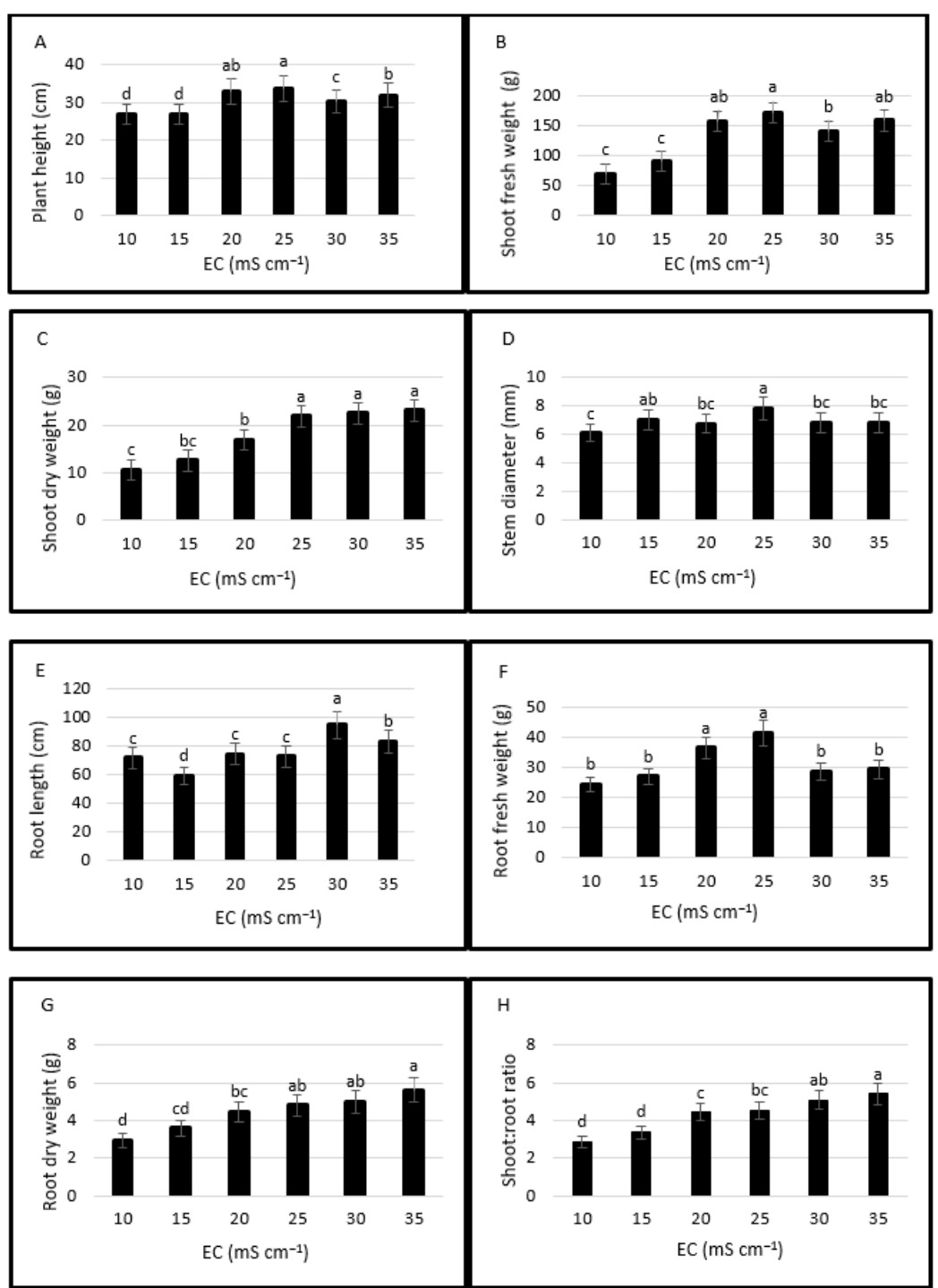

**Figure 3.** Effects of salinity levels on plant growth parameters (plant height (**A**), shoot fresh weight (**B**), shoot dry weight (**C**), stem diameter (**D**), root length (**E**), root fresh weight (**F**), root dry weight (**G**), shoot: root ratio (**H**). Error bars represent the standard deviation of the mean. Different letters mean statistically significant difference at *p* < 0.05.

### 3.2. Fresh Biomass

The effects of different EC levels of nutrient solution on measured fresh biomass were observed to be significant. The highest fresh biomass was 2864.06 g m$^{-2}$ at the EC level of 25 mS cm$^{-1}$ and followed by 2703.56 and 2457.15 g m$^{-2}$ at 35 and 20 mS cm$^{-1}$, respectively. The yield was 100.5 and 80% higher at 25 mS cm$^{-1}$ compared to 10 and 15 mS cm$^{-1}$,

respectively (Figure 4). It was 16.6, 26.3 and 5.9% higher at 25 mS cm$^{-1}$ than at 20, 30 and 35 mS cm$^{-1}$, respectively.

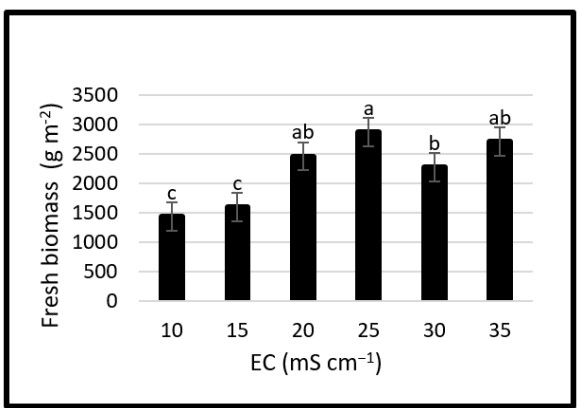

**Figure 4.** Effects of salinity levels on fresh biomass. Error bars represent the standard deviation of the mean. Different letters mean statistically significant difference at $p < 0.05$.

For leafy vegetables, yield potential greatly depends on biomass accumulation. The fresh biomass production of some halophytic species grown in hydroponics is affected by growing conditions, plant density, and salinity levels of the nutrient solution [66]. Biomass is significantly and positively correlated with the EC level of the growing medium [67]. Ventura et al. [25] reported that the total yield of two annual Salicornia and two perennial Sarcocornia ecotypes decreased with the increase of seawater rate above 50% and the cumulative yields of Salicornia types were between 13.4 and 16.04 kg m$^{-2}$ after 6 harvests until flowering while Sarcocornia types skipping the reproductive stage produced more (20.04 and 28.43 kg m$^{-2}$) in a year. Seedling establishment of *S. dolichostachya* among the 5 salinity regimes (0, 100, 200, 400 and 700 mM NaCl) was optimum in 100 mM based on the biomass increase, morphology, and taste [50]. Plant growth of *S. rubra* was optimum at 200 mM NaCl among the salinity levels of 0, 200, 400, 600, 800, and 1000 mM NaCl [68]. Plant growth of *S. bigelovii* reduced in 6 to 10 vs. 200 mM NaCl [69]. Shoot fresh weight of *S. europaea* had no phytotoxicity or deficiency at 100 and 300 mM NaCl while plant growth was restricted at 0, 500 and 700 mM NaCl [70], whereas *S. europaea* tolerated salinity levels up to 500 mM NaCl; however, plant growth performance was better in moderate salinities [71].

Boni [72] tested 2 (0 and 30 g L$^{-1}$) and 4 (0, 10, 20 and 30 g L$^{-1}$) salt concentrations in 2 experiments, respectively, and concluded that the fresh yield was above 7 kg m$^{-2}$ in less than 5 weeks up to salinity levels of 20 g L$^{-1}$. Not only yield values but also plant growth parameters impact the yield of Salicornia species. In an experiment conducted with high-yielding Salicornia genotypes at Marine Environment Research Center (United Arab Emirates), fresh shoots harvested twice yielded up to 23.7 t ha$^{-1}$ when the aquaculture effluents were used [63]. *S. brachiata* above-ground biomass varied in sampling sites and the highest biomass was observed in March ranging from 0.73 to 4.89 t ha$^{-1}$ [73]. Comparing our yield results with previous studies, the requisite salinity level at the root zone is similar to previous studies. However, in terms of fresh biomass, there are either similar or higher/lower ones closely related to growing conditions, length of production, harvested plant size, number of harvests, and whether they are from field or hydroponics.

### 3.3. Evapotranspiration

Evaporation is a combination of transpiration and evaporation. However, it is not easy to distinguish since both processes go together [74]. In floating systems, the amount of water loss through evaporation could be negligible because the ponds were covered with Styrofoam panels [75]. In this study, evapotranspiration altered between 6.89 and 10.9 L plant$^{-1}$. The lowest and the highest ET were at 10 (6.89 L plant$^{-1}$) and 25 (10.9 L plant$^{-1}$) mS cm$^{-1}$, respectively.

Evapotranspiration was lower at 36.79, 26.42, 22.75, 20.73 and 14.95% at 10, 15, 20, 30 and 35 mS cm$^{-1}$, respectively, compared with 25 mS cm$^{-1}$ having the highest value (Table 2). The highest ET is related to higher yield and plant growth values at 25 mS cm$^{-1}$. WUE was lower in the lower EC levels (10 and 15 mS cm$^{-1}$) having also lower fresh biomass. Higher WUE is related to better plant growth and higher fresh biomass [69,76].

**Table 2.** Evapotranspiration (L plant$^{-1}$) and WUE (kg m$^{-3}$).

| Treatment (mS cm$^{-1}$) | Evapotranspiration | WUE |
|---|---|---|
| 10 | 6.89 | 11.71 [b] |
| 15 | 8.02 | 11.2 [b] |
| 20 | 8.42 | 16.49 [a] |
| 25 | 10.90 | 14.84 [ab] |
| 30 | 8.64 | 14.82 [ab] |
| 35 | 9.27 | 16.49 [a] |
| Significance | | 0.020 * |

* Values followed by different letters are significantly different, $p < 0.05$.

### 3.4. Color

In this study, there were no significant differences in terms of L*, a*, and h° values among treatments whereas b* and C* values were found to be statistically significant (Table 3). In shoots of *Salicornia perennis* Mill., the b* value, which indicates a blue–yellow spectrum from −60 (blue) to +60 (yellow), was the highest (15.51) in the lowest EC treatment (10 mS cm$^{-1}$). This shows that the yellow color tone was more intense in the shoots of those plants. Similarly, the C* value, which expresses the intensity or color saturation of the shoots, was the highest (15.67) at 10 mS cm$^{-1}$, indicating that these shoots were brighter. L*, a*, and h° values of shoots of *Salicornia perennis* Mill. varied between 30.56 and 32.94, −2.42 and −1.79, and 96.95 and 105.77, respectively. These results are consistent with a report that the L*, C* and h° values of *Salicornia perennis* Mill. shoots were 31, 20.6 and 117.1 in May and 30, 17.3 and 123.2 in July at harvest [39]. Changes in color are associated with changes in chlorophyll content and carotenoids with increasing salinity [45,77].

**Table 3.** Shoot color analysis.

| Treatment (mS cm$^{-1}$) | L* | a* | b* | C* | h° |
|---|---|---|---|---|---|
| 10 | 32.28 | −1.84 | 15.51 [a] | 15.67 [a] | 96.95 |
| 15 | 31.76 | −2.42 | 13.74 [ab] | 14.03 [ab] | 100.18 |
| 20 | 30.56 | −1.97 | 12.08 [b] | 12.36 [b] | 105.77 |
| 25 | 32.09 | −1.79 | 14.36 [ab] | 14.51 [ab] | 97.60 |
| 30 | 31.24 | −2.36 | 12.38 [b] | 12.77 [b] | 101.75 |
| 35 | 32.94 | −1.89 | 13.61 [ab] | 13.90 [ab] | 105.04 |
| Significance * | n.s. | n.s. | *** | *** | n.s. |

n.s., *, *** refer to not-significant or significant at $p < 0.05$ and $p < 0.001$, respectively. Statistically significant means were subjected to Duncan's Multiple Range Test (MRT) at $\alpha = 0.05$. Means followed by different letters in each column are significantly different, $p < 0.05$.

### 3.5. Shoot Mineral Content

The effects of different EC levels of nutrient solution on shoot mineral content of *S. perennis* Mill. were found to be significant (Table 4a,b). Na concentration of the glassworts was in the range of 10.34–17.53%. Although the Na values were in the same statistical group over 15 mS cm$^{-1}$, they were 4.85, 3.29 and 10.23% lower at 35 mS cm$^{-1}$ than at 30, 25, and 20 mS cm$^{-1}$, respectively (Table 4a). The percentage of Na in dry weight increased as EC levels increased. The obtained values were found to be compatible with previous studies [39,78,79]. Cl content was also high in increasing EC levels. The accumulation of NaCl in the shoots is ta characteristic of Salicornia as a halophyte adjusting osmotic

potential and the Na content of shoots is associated with the concentration in the root zone [35,70].

**Table 4.** (**a**) Na, K, Cl contents and salt concentrations of *Salicornia perennis* Mill. at different EC levels. (**b**) Shoot mineral contents of *Salicornia perennis* Mill. at different EC levels.

| | | | | | |
|---|---|---|---|---|---|
| | | | **(a)** | | |
| Treatment (mS cm$^{-1}$) | Na (%) | K (%) | Na$^+$/K$^+$ Ratio | Cl (%) | Salt Concentrations (%, Dry Sample) |
| 10 | 10.34 [b] | 4.87 [a] | 2.12 [c] | 14.20 [e] | 23.40 [e] |
| 15 | 10.95 [b] | 4.43 [b] | 2.48 [c] | 15.98 [d] | 26.33 [d] |
| 20 | 15.74 [a] | 4.30 [bc] | 3.67 [b] | 23.79 [c] | 39.20 [c] |
| 25 | 16.95 [a] | 4.13 [cd] | 4.10 [ab] | 31.60 [b] | 52.07 [b] |
| 30 | 16.68 [a] | 4.11 [cd] | 4.06 [ab] | 32.31 [ab] | 53.24 [b] |
| 35 | 17.53 [a] | 3.97 [d] | 4.42 [a] | 33.73 [a] | 55.58 [a] |
| Significance | *** | *** | *** | *** | *** |
| | | | **(b)** | | |
| Treatment (mS cm$^{-1}$) | Mg (mg kg$^{-1}$) | P (mg kg$^{-1}$) | Ca (mg kg$^{-1}$) | Fe (mg kg$^{-1}$) | Mn (mg kg$^{-1}$) | Zn (mg kg$^{-1}$) |
| 10 | 5833 ± 27.5 [a] | 1941 ± 34.3 [a] | 1683.18 [f] | 68.1 ± 0.65 [a] | 72.42 ± 0.301 [a] | 41.09 ± 0.968 [a] |
| 15 | 5231 ± 39.78 [b] | 1776 ± 21.14 [b] | 1848.90 [d] | 61.04 ± 0.831 [b] | 47.84 ± 0.445 [d] | 29.07 ± 0.243 [c] |
| 20 | 4873 ± 24.01 [c] | 1538 ± 75.98 [c] | 2037.54 [a] | 46.64 ± 0.565 [e] | 61.39 ± 0.755 [b] | 27.61 ± 0.233 [c] |
| 25 | 4016 ± 9.64 [e] | 1418 ± 61.42 [d] | 2001.50 [b] | 52.47 ± 0.628 [d] | 73.04 ± 0.346 [a] | 32.34 ± 0.352 [b] |
| 30 | 4236 ± 14.09 [d] | 1537 ± 33.1 [c] | 1825.95 [e] | 55.83 ± 0.575 [c] | 47.61 ± 0.134 [d] | 23.94 ± 0.455 [d] |
| 35 | 3559 ± 13.06 [f] | 1318 ± 56.92 [e] | 1869.09 [c] | 47.69 ± 0.465 [e] | 51.46 ± 0.135 [c] | 25 ± 0.25 [d] |
| Significance | *** | *** | *** | *** | *** | *** |

Data are subjected to Duncan's Multiple Range Test (MRT) at $\alpha = 0.05$., *** refer significant at $p < 0.05$ and $p < 0.001$, respectively. Values followed by different letters are significantly different, $p < 0.05$.

The K value was between 4.87 and 3.97% at the EC value of 35 and 10 mS cm$^{-1}$, respectively. K content decreased as the EC value increased in particular at the EC level of 20 mS cm$^{-1}$. (Table 4a). As sodium uptake increased, potassium consumption decreased [80]. The Na$^+$/K$^+$ ratio was 2.12, 2.48, 3.67, 4.10, 4.06 and 4.42 at 10, 15, 20, 25, 30 and 35 mS cm$^{-1}$, respectively. A low Na$^+$/K$^+$ ratio is nutritionally critical because diets with high Na$^+$/K$^+$ ratios have been associated with the incidence of hypertension. In our study, the Na$^+$/K$^+$ ratio of the glassworts was in the range of 2.12–4.42. Altay et al. [79] reported the Na$^+$/K$^+$ ratio of the glassworts in the range of 5.64–10.15 which could be related to the origin of species collected from nature. The salt concentration values fluctuated between 55.58 and 23.40% in the EC value of 35 and 10 mS cm$^{-1}$, respectively; salt concentration values increased as the EC value increased (Table 4a).

Cl concentration was lower at 57.9, 52.62, 29.47, 6.31 and 4.21% at 10, 15, 20, 25, and 30 mS cm$^{-1}$, respectively, compared with 35 mS cm$^{-1}$ having the highest value (Table 4a). As the EC level increased, the Cl content increased [31,35]. The salt concentration values fluctuated between 55.58 and 23.40% at the EC value of 35 and 10 mS cm$^{-1}$, respectively. Salt concentration values increased as the EC value increased (Table 4a).

Mg, P, Fe, Mn, and Zn were the highest at 10 mS cm$^{-1}$, which was the lowest tested EC treatment. P and Mg contents displayed a similar response and decreased with the increasing salt concentrations. Ca content was the lowest at 10 mS cm$^{-1}$ and the highest at 20 mS cm$^{-1}$. Although there were slight changes in statistical groups, Fe, Mn, and Zn also indicated a decreasing trend with the increase of EC levels (Table 4b).

As the NaCl concentration increased, plants' magnesium uptake decreased. Magnesium is commonly found in significant amounts in all green vegetables due to its relationship with chlorophyll [81]. The Mg concentration value was lower at 10.32, 16.46, 31.15, 27.38, and 38.99% at 15, 20, 25, 30, and 35 mS cm$^{-1}$, respectively, compared with 10 mS cm$^{-1}$

(5833 mg kg$^{-1}$) having the highest value (Table 4b). As reported by Altay et al. [79], the Mg concentration of glasswort species was in the range of 680–15,700 mg kg$^{-1}$.

In our study, P concentration was lower at 8.5, 20.76, 26.94, 20.81, and 32.10% at 15, 20, 25, 30, and 35 mS cm$^{-1}$, respectively, compared with 10 mS cm$^{-1}$ (1941 mg kg$^{-1}$) having the highest value (Table 4b). Antunes et al. [39] reported the P content of *Sarcocornia perennis* as 2200.1 mg kg$^{-1}$. As the salt increased, the P content was found to be lower, which is in harmony with salinity studies conducted with different vegetables [82].

Ca concentration was in the range of 1683.18 −2037.54 mg kg$^{-1}$. Shoot Ca content was 21.05, 10.20, 1.80 11.59, and 9.01% higher at 10,15, 25, 30, and 35 mS cm$^{-1}$, respectively compared with 20 mS cm$^{-1}$ (Table 4b). Due to the difference in salinity levels of the salt marshes that the plants were collected from, these were observed to be lower than Lopes et al. [81] and Antunes et al. [39].

The highest Fe content was 68.1 mg kg$^{-1}$ at an EC level of 10 mS cm$^{-1}$ and followed by 15 mS cm$^{-1}$ at 61.04 mg kg$^{-1}$ (Table 4b). In our study, the Fe content of *Salicornia perennis* Mill. was detected to be lower than *Salicornia europea* reported by Al-Jaloud et al. [78] while it was higher than *Salicornia ramosissima* published by Lopes et al. [81]. The highest Mn content was 73.04 mg kg$^{-1}$ in the EC level of 25 mS cm$^{-1}$ (Table 4b). Although the Mn values were in the same statistical group, they were 0.85% higher in 25 mS cm$^{-1}$ than in 10 mS cm$^{-1}$. The Mn element content of *Salicornia perennis* Mill. was found to be higher than *Salicornia europea* [78]. In our study, the Zn concentration varied between 23.94 mg kg$^{-1}$ (the lowest in 30 mS cm$^{-1}$) and 41.09 mg kg$^{-1}$ (the highest in 10 mS cm$^{-1}$) and these observations were in harmony with previous studies. Antunes et al. [39] reported the Zn content of *Sarcocornia perennis* and *Salicornia ramosissima* as 21.2 mg kg$^{-1}$ and 33.9 mg kg$^{-1}$ respectively. Al-Jaloud et al. [78] reported the mean Zn concentration of *Salicornia europea* as 24.48 mg kg$^{-1}$.

In the study, the differences in EC levels of the nutrient solution changed between +0.6 and +1 mS cm$^{-1}$; however, shoot mineral contents were in harmony with previous studies although nutrient elements decreased with increasing EC levels excluding Na and Cl.

## 4. Conclusions

This study, conducted with *Salicornia perennis* Mill., has demonstrated that glasswort plants as a new crop in floating systems have a potential for commercial production considering the yield performance without any nutrient deficiency. The yield was higher at moderate EC level of nutrient solution, and it was concluded that the EC level of nutrient solution could be adjusted to 25 mS cm$^{-1}$ with NaCl for hydroponic production. However, more research is needed on how plant density, harvest numbers, and growing conditions affect the yield.

**Author Contributions:** Conceptualization, E.O. and Y.T.; methodology, E.O. and Y.T.; software, E.O. and Y.T.; investigation, E.O. and Y.T.; resources, E.O. and Y.T.; writing—original draft preparation, E.O. and Y.T.; writing—review and editing, E.O. and Y.T.; visualization, E.O. and Y.T.; supervision, E.O. and Y.T. All authors have read and agreed to the published version of the manuscript.

**Funding:** This research received no external funding.

**Institutional Review Board Statement:** Not applicable.

**Informed Consent Statement:** Not applicable.

**Data Availability Statement:** The data presented in this study are available on request from the corresponding author.

**Conflicts of Interest:** The authors declare no conflict of interest.

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
