# Peer review of "Effect of EC Levels of Nutrient Solution on Glasswort (Salicornia perennis Mill.) Production in Floating System"

_horticulturae, doi:10.3390/horticulturae9050555_

Round 1
Author Response
Lütfen eke bakınız.

Reviewer 2 Report
All comments are inserted in document attached.

Author Response
Lütfen eke bakınız.

Round 2
Reviewer 1 Report
I have read the revised manuscript. I think that the results are not well discussed. There seem to be too many references cited. Please interpret the data from this experiment.
The authors should also correct the points listed below.
L23
Soilless agriculture → Soilless culture
L23
Salicornia perennis : Please change it to another word.
I think that the keywords would be words that are not included in the title.
L105, L122
Please also note that the film in the greenhouse blew away in the wind. When did you put up the tunnel? Write down the exact conditions of the experiment.
L115
Please write down the numerical values of oxygen concentration.
L134
I apologize for missing the Na and Cl data.
L210, L237
Bars with different letters represent a significant difference.
→Error bars represent the standard deviation of the mean. Different letters meas statictically significant difference at p < 0.05.
L176, L278, L388-390, L394
p →p  Change to italics.
L393
α →α  Change to italics.
Table 4b
Please align the digits of the mean and SD values.
L177-385
The authors describe the values for each element in detail and compare them to previous reports. I know the amount of growth and the concentration of components in this period of time. As a reader, I would like to know if that makes Glasswort worthy of being produced as a new crop. Please add your “discussion”. Is there a target or guideline FW, mineral concentration?
Author Response
Lütfen eke bakınız.

Reviewer 3 Report
1st previous review comments
1. It is necessary to describe the conditions of the sampling site (EC, pH, moisture content, etc.). Fresh weight information is also required.
2. Specific explanation is needed for the replicates.
3. Figs: Explanation of the error bars is required.
4. It is necessary to separate the amount of transpiration and the amount of evaporation.
2nd review comments based on the 1st comments above
1. It is still necessary to describe the conditions of the sampling site (EC, pH, moisture content, etc.) because the growing environment to which the seedling material is adapted often greatly affects the experimental results.
2. Specific explanation is still needed for the replicates. For example, it should be clear if the experiment was replicated 3 times at different times or if 3 culture beds were used for each treatment.
3. Figs: Explanation of the error bars is still required. Please specify whether the error bars are standard deviations or standard errors. The ± values in the table also require a similar explanation.
4. It is still necessary to separate the amount of transpiration and the amount of evaporation because evapotranspiration is highly dependent on the cultivation system. Readers are expecting to learn about the transpiration rate of plants and WUE findings based on it under different conditions. For more scientific knowledge, the transpiration rate should be obtained by subtracting the evaporation value (or its estimated value) from the measured evapotranspiration value of the system.
Author Response
Lütfen eke bakınız.
